# Education and Licensing of Horse Owners: Addressing Poor Horse Welfare in the UK

**DOI:** 10.3390/ani15071037

**Published:** 2025-04-03

**Authors:** Aurelia Hall-Bromley, Laura Dixon

**Affiliations:** Animal Welfare Centre, The Royal (Dick) School of Veterinary Studies, Easter Bush Campus, The University of Edinburgh, Midlothian EH25 9RG, UK; aureliahall@hotmail.co.uk

**Keywords:** horse welfare, education, licensing, equine

## Abstract

Horse welfare concerns have been a subject of increasing discussion for decades. The issue of compromised horse welfare in the UK and across the globe is complex as welfare concerns are varied, wide-ranging and nuanced. Several solutions to combat compromised horse welfare have been suggested by existing research; however, each of these has its limitations. In this study, existing literature regarding horse welfare, education and licensing as potential solutions were analysed. Distress behaviour was the most-cited welfare concern in the literature, narrowly followed by health issues and behavioural issues. The citations for causes of poor welfare were dominated by management and training practices. The analysis found that the highest cited barrier to good welfare was a lack of knowledge, followed by a lack of understanding by horse owners. Further field research into horse welfare causes was most commonly suggested as the best step to address welfare concerns, and increased awareness of welfare issues was suggested as the best solution to prevent welfare issues. In terms of education, the most-cited positive outcomes were increased knowledge, awareness or understanding. However, the most prominent limitation of education was an unclear effect on behaviour as well as other notable factors, such as availability. The most-cited licensing success was used as a consumer tool to aid decision making; however, licensing was limited by enforcement. Taking the relevant literature as a whole, there is no single evident solution that can solve the horse welfare problem; however, there are areas identified that merit closer consideration.

## 1. Introduction

The role of the horse in human society has evolved over the years, with a shift in recent decades towards the position of sporting or companion animals in the developed world, including the United Kingdom [1,2,3]. Any use of horses includes a range of activities and interventions that can, collectively, compromise welfare [4,5]. Domestication itself has been identified as a source of welfare issues, for example, through restriction of movement, social interaction and equine behaviour patterns [1]. Subsequently, recreational horse welfare has become a prominent global concern due to the lack of regulation, as well as the increased diversity of horses and owners, leaving more room for knowledge gaps [6]. Additionally, the popularity of equestrian sport has placed horse welfare under increased scrutiny [7], with ongoing work to reduce risk to an acceptable level [8]. Animal science has also advanced, with welfare science developing rapidly alongside the evolution of the concept of welfare itself [9]. Finally, there has been an emergence of equitation science, which enables us to reconsider horse training practices using learning theory, physics and ethology, with the goal of achieving optimal training aims without compromising welfare through fear or pain [10].

Despite the history of discussion and research demonstrating avoidable risks to the welfare of horses and methods for prevention, recent findings indicate that there remains a high prevalence of welfare problems for horses [11] caused by certain training techniques [1] and husbandry practices [7]. For purposes of this review, the term “welfare” is defined as an individual’s long-term physical and mental state and perception of their situation [12], with consideration of the needs of animals, as stated by the Animal Welfare Act 2006, and their interplay according to the Five Domains Model [13]. For the United Kingdom (UK), estimates of the size of the horse population vary. DEFRA estimated 150,000 horses—in commercial holdings—in 2021, the UN estimated 425,000 in 2019 [14], and the British Equestrian Trade Association (BETA) estimated 847,000 in the same year [15]. BETA also estimated 374,000 horse-owning households in Britain. The limited estimate for the UK horse population is a concern in itself despite requirements for equine passports and microchipping; however, these figures serve to demonstrate the scale of the issues at hand. Almost 850,000 horses are potentially at risk of welfare compromise, with less than half as many owners to take responsibility. Due to the number of privately owned horses and lack of any tailored government legislation specifically for their ownership or management, this review focused on their welfare, with horses involved in the reviewed research being predominantly recreational, sporting or companion animals, as opposed to commercially owned or managed horses.

In 2022, new legislation was introduced in France to make the education of horse owners mandatory through licensing [16]. Although it is too early to assess the success of this legislation, this raises a thought-provoking question. Could education and licensing of horse owners improve the welfare status of the UK horse population? The improvement of welfare issues for UK horses has been described as a relentless challenge due to the complexity of the issue and public perception [17]. Monitoring and reporting also appear to be an issue; not only is the size of the population poorly estimated, but there is a limited overview of the population’s welfare status [18], although attempts have been made to assess the prevalence of specific concerns [18,19,20]. To determine if education and licensing could be a potential UK solution, first, the scale of horse welfare concerns needs to be determined, the likely causes for poor welfare identified, the existing suggestions for improvement explored and the possible applications and limitations of education and licensing assessed.

Additionally, in order to identify a possible successful solution to compromised horse welfare on a national scale, we need to understand the priorities and values held by key stakeholders [21]. For recreational horses, welfare is primarily the responsibility of the owner [7], making the owner the primary stakeholder. Many recreational horse owners have little or no previous knowledge of horse needs [1]. The need for improvement of horse-owner knowledge has been echoed in relation to a variety of welfare concerns [2,22,23,24]. Also, studies have found horse welfare can be impacted by different facets of the horse–human relationship [25,26,27]. Research aimed at understanding stakeholders’ perceptions of horse welfare has been limited [28], as has research into horse–owner relationships [7]. However, several common themes have been identified as factors contributing towards poor horse welfare: accepted social norms, ignorance, indifference, financial constraints and indolence [18,24,29]. Each suggested factor can fall under capability (C), opportunity (O) or motivation (M); the suggested processes underlying human behaviour (B) in the COM-B model [30], which can be targeted to achieve effective behaviour change [31]. Achievable human behaviour changes need to be considered for effective legislation to be developed, followed and enforced.

The aim of this review was to determine if structured education and owner-specific legislation could be a potential solution to common welfare issues of privately owned horses. To meet this aim, the review distilled the most studied welfare concerns relevant to the privately owned UK horse population within the existing research literature by assessing the frequency of research into different welfare issues across the developed world from countries with similar social, political and economic status to the UK and comparing the findings with existing research focussed on the UK. The reasons behind continued welfare issues were also assessed through analysis of poor welfare causes and barriers to good welfare and these were also considered alongside successes and limitations of education use. Finally, the review considered licensing as an enforcer of education and a regulatory tool for horse ownership and welfare issues by assessing and summarising the cited successes and limitations of legislation within the existing literature.

## 2. Materials and Methods

This study was a systemised literature review using the principles of a systematic literature review on a smaller scale. The data were collected between January and March 2023, and the findings assessed between March and July 2023. The review was guided by the PRISMA protocol [32] to enable accurate reporting [33]; the recently updated reporting tool was used [34]. See Appendix A for the completed protocol for this project. The PRISMA checklist was used to ensure that the review had clear aims and objectives and a planned methodology prior to data collection. Any amendments to the methodology are specified, as well as the elements of a full systematic review, which were not possible in this study due to the limited time frame.

The review used two discrete searches to encompass both the horse welfare literature (Search 1) and the literature exploring education and licensing in relation to animal welfare (Search 2). The searches were run first in Web of Science and cross-checked in Google Scholar for comprehensive coverage.

Both searches were limited to the date range of 2006–2023, from the latest update to the Animal Welfare Act, which altered the standards of and obligation to welfare for horse owners [35,36], until the start date of data collection. The literature had to be cited at least once a year on average since publication, i.e., ten times in ten years or twice in two years. This criterion reduced time spent critically analysing papers that were not taken up by continuing research in the field, which may indicate less robust or practically applicable studies or may have been superseded by newer research, thus maintaining the scope of the study within the available timeframe. Both searches excluded literature focussed on developing or least developed countries according to the World Trade Organisation [37] and the classification by the United Nations [38] in order to obtain results most relevant to the UK’s social, economic and political climate. Search 1 was also subject to the following additional exclusion criteria: settings not relevant to private ownerships (for example, veterinary procedures, research, breeding, racing, meat production and industrial operations). Additionally, papers exploring new and alternative methods for assessing welfare were excluded, as their focus was on practical welfare indicators rather than the concerns themselves or accompanying factors. Search 2 also excluded literature focussed on veterinarians or with an industrial focus, as research in these areas was not relevant to licensing or educating the general population.

Table 1 provides details on the search terms applied. Both search terms yielded several thousand results in Web of Science. An additional criterion was added to the results to include “animal welfare” in the topic (title, abstract and keywords). To condense the sample down to the most relevant items and for them to be assessable within the scope of this project, further exclusions were applied. Specifically, the searches were then limited to articles, review articles and entries in English (the majority of results). Search 2 also excluded papers with citation topics of veterinary sciences, nursing and marine biology to automatically exclude papers according to the predetermined criteria. All remaining papers were recorded and checked for potential relevance to the data set, and any not meeting the criteria were manually excluded. Google Scholar did not allow for as detailed filtering of results as the Web of Science. To enable a cross-check within the scope of this review, this search was limited to essential keywords within the title, using the Advanced Search function.

Every paper that met the inclusion criteria was analysed for the validity, repeatability and generalisability of the methods and results. For each paper, the type of methodology and the sample size were recorded, with comments on any potential for bias and observations of the robustness of the methods and quality of the sample.

Descriptive statistics were employed to analyse the data and enable inferences for wider discussion [39]. Frequency distribution and mode were of primary interest, as they would identify which topics were most and least researched or discussed in the literature. While this methodology does mean that the scale of welfare concerns is somewhat reliant on what is easiest or most popular to research, the frequency of papers under each category still served to provide a general picture of the concerns faced in the 17 years preceding this study. Responses were collated under several themes. Search 1 was analysed for details relating to welfare concerns, barriers to good welfare, causes of poor welfare, and suggested solutions/next steps. Search 2 was analysed for details relating to successes of licensing, successes of education, limitations of licensing, and limitations of education. Each paper within a search could contribute multiple inputs across the available themes; for example, in Search 1, a paper could discuss multiple welfare concerns, such as pain, distress and coercive control, while also identifying possible causes, such as training practices and competition. The total frequency of inputs under each theme was also recorded. Measuring the frequency of welfare concerns within the literature could not accurately determine the prevalence or ranking of those concerns for the UK horse population. However, the results were searched for studies that assessed the prevalence, priority, causes and solutions of horse welfare concerns in the UK, and their results are compared to this study’s findings. This can illustrate whether research is currently reflecting the perceived issues in the sector and, thus, whether conclusions from the research are more likely to be applicable to welfare improvement strategies.

## 3. Results

Overall, both searches yielded more applicable papers than anticipated, generating a wealth of data across many themes. It should be noted that the themes for both education and licensing successes/limitations include both influential and outcome factors; for example, Education Successes include parameters that have been argued to make education successful and outcomes that are assessed as a beneficial result of education. Table 2 summarises key findings, the number of papers that fell under each theme (input frequency) and the number of different categories identified within each theme (input variation). Appendix B defines each theme and input in the context of this study, while Appendix C contains the full list of included papers.

A variety of methods and samples were present in both data sets. However, a common theme, with few exceptions, was a gender skew in studies involving human participants. Generally, these studies had more female participants than male, some with as much as an 85:15% split, respectively. This limitation of the literature with a view to informing future action will be discussed further. Studies in Search 1 had numerically smaller sample sizes, on average, than those in Search 2. This is likely due to the larger number of studies involving animals where very large sample sizes can be difficult to obtain, as opposed to sample sizes involving human participants, as in Search 1., which should not affect the validity across the two data sets.

### 3.1. Search 1

Search 1 produced 933 results (413 from Web of Science, 520 from Google Scholar), yielding 124 included papers (89 from Web of Science, 35 additional from Google Scholar). These included 43 Quantitative Experimental studies, 18 Qualitative (interviews, focus groups, etc.), 28 Surveys (quantitative data) and 35 Literature Reviews. Many of the excluded papers were focussing on developing welfare assessments, as well as a small number of studies from developing countries.

### 3.2. Search 2

Search 2 produced 605 results (455 from Web of Science, 150 from Google Scholar), yielding 135 included papers (130 from Web of Science, five additional from Google Scholar). These included 15 Quantitative studies, 27 Qualitative, 68 Surveys and 25 Literature Reviews. Of these, eight were also included in Search 1. Most excluded papers were due to a veterinary or industrial focus.

### 3.3. Descriptive Statistics

The frequency distribution for each theme is presented in Figure 1, Figure 2, Figure 3, Figure 4, Figure 5, Figure 6, Figure 7 and Figure 8. The modal result for each theme is presented in Table 3.

Distress (including stress and fear) was the most-cited welfare concern in the literature, narrowly followed by health issues (including disease) and unwanted behaviours (including stereotypic, aggressive and hyperactive behaviours) (Figure 1). Also highly cited were confinement, pain and isolation (Figure 1).

The cited causes of poor welfare were dominated by management practices and, to a lesser extent, training practices (Figure 2). There are other notable causes, which will be discussed in more detail. However, many of these can also be related to these two dominant causes; for example, Human–Animal Interactions (HAIs) are intrinsically part of most management and training practices. The highest cited barrier to good welfare was a lack of knowledge, followed by a lack of understanding, of available information about horses’ needs and personal beliefs (Figure 3).

The most-cited solution/next step to improve horse welfare was further research; this was another dominant theme alongside increased communications and awareness (Figure 4). However, three other themes stood out in the data set, albeit to a lesser extent: culture/behaviour change, education and industry-led initiatives (Figure 4).

The successes of education were another data set dominated by increasing understanding, knowledge or awareness, with the secondary success of education indicated as aiding cultural change. One of the highest cited factors for education to be successful was institutional/national support (Figure 5); the possible interplay of this with the suggested solution of industry-led commitment (Figure 4) will be discussed in more detail later. The cited limitations of education were most prominently an unclear effect on behaviour (human); however, availability, other circumstances and syllabus content were also notable (Figure 6).

There were less data present in the data set for licensing successes and limitations. However, from the available inputs, the most-cited success of licensing was as a consumer tool—e.g., license information in packaging to inform consumers aid decision making (Figure 7)—and the most-cited limitation was enforcement, followed closely by available resource for implementation (Figure 8). There are other moderately cited factors, such as monitoring and regulation as a success of licensing (Figure 7), which will be discussed further.

## 4. Discussion

The variety of welfare concerns cited in the literature (Table 2, Figure 1) emphasises the complexity of welfare as an issue. The results of this review will be assessed in relation to the prevalence of welfare concerns and their corresponding causes. Additionally, reasons for and against education and licensing as a method to improve privately owned horse welfare will be evaluated with consideration of the cited barriers to improvement.

Although further research is the most suggested next step to address horse welfare concerns in the assessed literature (Table 3), this discussion is primarily focused on how existing knowledge could be used to combat horse welfare issues now, which will make up the majority of the evaluation. There are undoubtedly areas where further research is needed, and those that are key to this study’s findings will be discussed. However, in general, more research itself will not necessarily provide solutions and may actually lead to more questions or areas of further research. This is still valuable; however, is less likely to be directly applicable in practical scenarios that private horse owners can implement

### 4.1. Welfare Concerns and Causes

Within the assessed literature, few papers have also sought to prioritise or rank horse welfare concerns in particular countries (e.g., [18,40,41]). Rioja-Lang et al. and Horseman et al. surveyed equine experts and/or stakeholders to determine UK horse welfare priorities. Overall, the highest-ranked concerns by Rioja-Lang et al. [41] and Horseman et al. [18] can be linked back to management practices that would appear to support the initial findings of this study (Figure 2). The differences between this study and those consultations and interviews could be for assorted reasons. Firstly, Rioja-Lang et al. [41] explored a consensus for a range of species as well as horses; thus, the sample size for horse welfare data was only 19 of the 117 experts consulted, compared with a sample of 31 by Horseman et al. [18], whereas this study included contributions from 124 papers. Although, the Horseman et al. [18] study used an opportunistic sampling technique, the data collected may not be as comprehensive as the more structured approach by Rioja Lang et al. [41]. Equally, the focused data collection of interviews and focus groups cannot be directly compared to the findings of a broad literature review. Secondly, the Rioja Lang et al. study [41] was only seeking expert opinions, primarily researchers, charity workers and industrial representatives, with small representation from vets or government workers. By contrast, the data set for this study is based on a wide range of research, including opinions from owners and vets. Thirdly, the study by Rioja-Lang et al. included concerns that were excluded from this study, such as euthanasia decisions and breeding. Finally, the Rioja-Lang et al. [41] study categorised some concerns differently. For example, fear/distress/injury resulting from specific practices compared to stress/distress/fear being one category and work, training, health concerns and pain being separate categories. Similarly “lack of recognition of pain behaviour” versus pain itself as a category and “large worm burdens” as a specific category that would be encompassed within health in this study. On the other hand, obesity is categorised the same in both studies, and some terms are very similar, such as “unsuitable diets” and feeding regimes, and “poorly fitting and restrictive tack” compared to simply tack (including bit use). The former study also limited the rankings to the top ten, whereas this study has counted all cited welfare issues. However, the similarity of results both in Rioja-Lang et al. [41] and Horseman et al. [18] concurs with this study’s findings, as management practices are the most-cited cause of poor welfare (Table 3). Management practices encompass some of the most-cited welfare concerns, for example health issues (e.g., [42]), and confinement and isolation (Figure 1) (e.g., [43,44]). Management practices can also include feeding regimes that themselves ranked within the top ten causes of welfare concerns (Figure 2) (e.g., [45]). As the horse owners for privately owned horses are mainly responsible for these practices [46,47], interventions aimed at private horse owners have the potential to improve welfare for a large population of horses.

Training practices are another area of concern, according to the literature (Figure 2). For example, it has been argued that lungeing, or round-pen training, poses a significant welfare concern without proper application of learning theory [23]. Horses may perceive themselves as being chased without a means of escape as opposed to being at liberty to respond to training stimuli [23]. This could lead to ‘learned helplessness’, whereby horses learn that movements to escape a negative situation are not possible, leading to poor welfare outcomes [11,48]. Equally, horses can fail to learn established boundaries or engage with training stimuli if these practices are undertaken improperly [9]. Training practices can also be linked to other highly cited welfare concerns of distress, unwanted behaviour, pain, injury and coercive control (Figure 1). The existing literature suggests a number of reasons that training practices may be a source of welfare concerns, even putting aside intentionally harsh practices that have been largely outlawed. Training practices may be hindered by misleading terminology that is poorly understood [49]. Equally, training may be limited by practices developed prior to recent findings. For example, research into horse cognition has yielded results that contradict commonly held beliefs about horse behaviour [50], including the formation of stereotypic behaviours in relation to dopamine regulation. These discoveries also, in turn, encourage reflection on common training and management practices, such as positive versus negative reinforcement learning techniques and the still widespread practice of long-term stabling (confinement) [51]. As a result, understanding horse learning can benefit welfare as well as performance [52]. Public outreach has been suggested as a method to apply scientific discovery to horse training, breaking down barriers and challenging long-held beliefs [51]. However, depending on individual beliefs, equestrians can be sceptical about science [53]. Without sensitivity to this, suggesting such solutions would only continue the apparent mismatch between theoretical knowledge and field applications [11]. While scientific advances can be communicated simply, there is currently no evidence to suggest that solely increasing awareness improves welfare assessment skills, although this may be a limitation of the study design rather than evidence of no effect [13]. Similarly, for management practices, a recent study delivering research-based information to horse owners via webinar found an increase in knowledge but was not able to assess whether the information was applied or skills increased [53]. Therefore, further research may be required to assess the impact of increased knowledge or awareness on horse welfare in practice. For example, research that analyses the delivery of welfare education could include or lead to a follow-up study that assesses the welfare of horses under the care of attendees and evidence of any changes in their management after the learning experience to provide evidence for or against this as a welfare improvement strategy.

### 4.2. Recent Developments in France

In France, the horse population has been expanding since 1995, reaching 950,000 in 2010 [54]. This growth is the result of the development of pony riding for children and the increasing interest of French people in recreational riding and horse betting [55]. Concerns in France are like those in the UK: human–horse relationships, economic efficiency, environmental issues, preservation of horse breeds, land-use pressures, and the health, welfare, and care of animals, including death [54].

In 2021, France introduced new legislation, to be brought into effect in 2022, that would require all horse owners to undertake education to demonstrate their knowledge about equine management [56]. This education strategy is different from the public outreach to raise awareness discussed above, as it has a more formal course and teaching structure with evidence of knowledge required. The goal of this legislation and education programme is to combat abuse, and the initiative has been lobbied for since the introduction of licensing for Federation Française D’Equitation (FFE) members in 2019 [16]. This presents an interesting idea for other countries experiencing persistent cases of compromised equine welfare. The French initiative would appear to target the leisure riding community, as competitive riders are already required to have a membership [57]. To be licensed, owners and riders have to complete the Gallop 4 qualification, or an equivalent, that covers general horse knowledge, horse care, equestrian practice and safety practices.

With the growing understanding of human behavioural science [58] in mind, consideration of horse-owner values and perceptions should inform the practicality and viability of applying a similar change in legislation in the UK. When applying human behaviour change principles to a specific horse welfare concern, obesity, it was found that environments and social norms limited the likelihood of proactive action by horse owners, and there were issues with owners’ ability to identify and assess such health issues or weigh up the short-term impacts of weight management against the long-term health benefits [59]. Although that study was focused on one specific area of horse welfare, it covered a significant sample size and effectively applied a mixed methods approach to obtain a holistic view of the issue; therefore, this method could be extrapolated to consider other welfare issues in turn. However, the questions asked in this study remain: can education, either voluntary or mandated through a licensing system, overcome barriers to human behaviour change and directly benefit animal welfare?

### 4.3. Education and Licensing in Principle

While this study aimed to explore both education and licensing with respect to their ability to improve horse welfare, less research was conducted on the successes and limitations of licensing than on education. Thus, there is a skew in the discussion and analysis towards the costs and benefits of education and barriers to this as a solution to horse welfare concerns. As education appears to be the primary objective of the French horse-owner license, and the results suggest that successful licensing relies upon education (Figure 8), it is reasonable that this discussion places a greater focus on education. However, licensing will be explored, with the potential for combined enforcement and independent regulation.

#### 4.3.1. Education and Horse Welfare

Education has the potential to make a difference in horses’ welfare through their owners; in fact, educating owners has been argued as combatting poor welfare in other domestic species [60,61]. The lack of knowledge is the highest-cited barrier to horse welfare (e.g., [42]), with a lack of understanding of the available information about horses’ needs being second (e.g., [62]) (Figure 3). Concurrently, increased knowledge, awareness and understanding were the most successful aspects of educational interventions (e.g., [63]) (Figure 5), with increased awareness being the second-ranked solution to poor horse welfare (e.g., [64]) while education was the fourth (e.g., [65]) (Figure 4); however, barriers to increasing knowledge and understanding are similar to those discussed above. Cultural change was the second-ranked education success (e.g., [66]) (Figure 5); however, the most common education limitation is an unclear effect on human behaviour, (e.g., [59]) (Figure 6). Cultural change can be shown in non-material forms (e.g., language), and this may explain its high ranking in education success; however, material changes (e.g., practical application) are needed for modifications that affect horse welfare. Therefore, this calls into question the usefulness of education in creating material culture/behaviour changes (third-ranked solution) as a method to improve horse welfare (e.g., [67]) (Figure 4).

To explore this further, we must look in more detail at other factors involved in horse welfare. Notably, amongst the causes of welfare concerns are the following: competition, attitudes/beliefs and owner perception, as well as limitations of facilities (Figure 2). The first three of these are within the owner’s control: what goals they set, what they believe is required or right to achieve their goals, and what they see as good or bad for welfare and performance. While education may influence the culture around these facets of horse welfare (Figure 5), the previously highlighted discrepancy calls into question whether it will lead to meaningful human behaviour change (Figure 6). Limited facilities are an interesting addition to the data set, as these could be beyond an owner’s control, such as within the context of a livery yard. Livery restrictions have been suggested as a barrier to lasting behavioural change following educational intervention [13]. Facility limitations have been associated with cited welfare concerns, such as behavioural problems [68], weight problems [44], confinement and isolation [49] or crowding (insufficient space) [69]. Overpopulation, cited as another cause of poor welfare (e.g., [41,68]), could exacerbate limited facilities; if there are too few well-equipped facilities for the horse and owner population, then owners’ choices become limited. Overall, facility limitations may also limit the success of an educational remedy to compromised horse welfare, unless other regulations are introduced [70]. Another challenge is that privately owned horses may be kept in a variety of environments aside from livery yards, such as the owner’s home property. Facility limitations of these environments may still exist but may also be different than those of horses housed in livery. Exploring the differences in welfare concerns and the potential opportunities and challenges of education and licensing to improve welfare between different populations of privately owned horses was out of the scope of this review but may provide interesting data in future research that could help tailor recommendations to specific populations.

The issue becomes more complex when considering that beliefs, resistance, lack of trust, and personal bias are also potential barriers to good welfare (e.g., [8]) (Figure 3). Additionally, financial constraints, circumstances, awareness, communications, and lack of clarity all may influence the ability of education to improve individual horse welfare [58]. The last three of these factors are also affected by the horse industry and the government, who have the ability to provide clear communications and ensure that prominent issues are transparently presented with any disparities between advice streams being resolved [70]. This would follow suggestions two and five of solutions to welfare concerns (Figure 4—increased communications and industry-led initiatives/commitment). However, wider education resources may be needed to ensure an understanding of complex issues by novice owners, as well as the uptake of new ideas [71]. The first three barriers mentioned above (beliefs, resistance/lack of trust, and personal bias) are also complicated as they are rooted in an individual’s culture, experience and previous education [51]. Each individual has their own perception of what good horse keeping entails [72]. This is where communications can still play a key role; using appropriate terminology and communication channels, trusted sources and trying to align messages within existing belief frameworks make it more likely that important messages will be positively received by horse owners [73]. Trusted sources are often known contacts or groups [45]; however, there may be scope for experts on specific welfare topics to raise awareness within these smaller communities and build trust with private horse owners [74]. On the other hand, financial constraints and other circumstances are challenging barriers to overcome, and it is significant that wider circumstances are highly ranked as a limitation of education (Figure 6). These restrictions are influenced, in turn, by a wide range of other factors, including the economy, work pressures, geographical location, life changes, and other individuals’ actions [61]. These circumstances can make it hard for theory to be put into practice; an owner can know the optimal way to care for their horse but hit a roadblock if they do not have enough time, or if the cost of essential horse maintenance increases. This barrier would require more than well-designed communications to be overcome, indicating that education can target some of the root challenges to good horse welfare; however, it cannot reduce non-horse-related external pressures on horse owners.

We should also look closely at other limitations of education. The unclear effect on human behaviour is the most problematic, as an overall change in attitude and behaviour towards more welfare-friendly practices is evidently sought after throughout the literature (Figure 4) [75]. In 2012, Visser and Van Wijk-Jansen [45] found that many participants had good awareness of issues compromising horse welfare; however, their knowledge was not consistent in producing appropriate practices. There could be many reasons for theory not translating into application by horse owners, including external factors. A particular barrier that has been suggested is the perception of the issues, which ranks fourth as an education limitation (Figure 6) (e.g., [76]). Thompson and Clarkson [73] suggested that there was a disparity between owner self-reporting and observations by a trained professional regarding horse behaviour. Furtado et al. [77] also cited that owners demonstrated inaccuracy in assessing their horse’s weight. The perception and prioritisation of issues are linked closely to personal beliefs and biases that may be barriers to welfare (Figure 3) [9]. There is scope for education to reduce this barrier, as education has been found to improve openness to new ideas (Figure 5) [78]. However, there is some ambiguity around this, which will be explored further.

Availability and syllabus content are ranked as the second and third limitations for education to be successful (Figure 6). Availability can be affected by other limitations, including resource (e.g., [78]) and resistance, alongside lack of uptake (e.g., [28]) (Figure 6). Making education available is challenging as it requires infrastructure; institutional/national support is the highest-ranked input for education to be successful (Figure 5) (e.g., [79,80]). Justification of resourcing is difficult if there is a lack of uptake, and stakeholders have expressed scepticism about whether those who really need education would be likely to volunteer for learning or be truly impacted by the information [27]. Although this belief may be biased within the small sample of the Collins et al. study [27], the concern would appear to be supported by the findings here, as education’s limitations include resistance and limited influence (Figure 7), as well as a limited effect on behaviour as discussed above. Uptake becomes less of a problem if education is made mandatory. The question is then about the likely effectiveness and the value of implementation. On the other hand, syllabus content can be controlled with sufficient collaboration between governing bodies, key stakeholders and education providers. A modern syllabus that is up to date with recent scientific findings and the range of currently held beliefs in the community could improve the appeal of voluntary education. The problem, then, is agreement over delivery. There is a lack of consensus in the literature over how best to deliver welfare education, with arguments being made for interactive [79,81], online [78,82], communication-based [83], animal-centred [84], multilevel [85], interdisciplinary [86,87] and assessed methods [84]. The idea of online learning also has merit. MacKay et al. [78] found that Massive Open Online Courses had a high success and acceptance rate throughout a large and diverse sample of participants. The COVID-19 pandemic also created an increase in testing and delivering online learning. Sasaki [82] found that practical horse training for veterinarians could be delivered online without compromising achievement, maintaining student satisfaction and increasing self-directed learning. While this was a pilot study, it had a straightforward and repeatable method that could be worth expanding for generalisability. The sample in the Sasaki study [82] was also well balanced in terms of participant gender, which is unusual amongst the assessed literature, which tends to be majority female, indicating a high potential accessibility or appeal of online learning. Further research or consultation would be required to ascertain which method of delivery would be optimal for horse welfare education.

There is a discrepancy in the results as to whether education improves individuals’ ability to respond to change. One of the goals for improving welfare is to move away from outdated ideas [10]. Ten papers cite being open to new ideas as an education success (Figure 5) (e.g., [88,89]); however, four papers cite reduced perspective taking as a limitation (Figure 6) (e.g., [90]). To determine whether education is a useful tool towards this goal, it needs to be discerned whether education is likely to enable horse owners to change their practices or if it may limit their perspective of the issues. As all the respective papers within the assessed literature are mainly focussed on meat production and consumer choice, further research is needed to determine specifically whether education can influence horse owners’ reception of new information.

There is also a question around improved skills, in this case, the ability to improve horse welfare through management and training practices. Studies have found correlations between human skills and animal welfare [90,91]. Improved skill is ranked as an education success by four papers in the assessed literature, (e.g., [91,92,93]) (Figure 5); however, one paper cited a lack of improved skill following education (Figure 6) [12]. The study by Fletcher et al. [12] did not find a difference in participants’ skills to detect welfare parameters based on their level of equine-related academic qualifications. This could be indicative of a lack of focus in the syllabus on welfare assessment or a lack of distinction in the syllabus regarding welfare definitions. It should also be noted that the sample size for that study was small compared to the representative population and would need expanding to truly evaluate any potential differences in educated and non-educated equestrians’ abilities. Again, the papers advocating for education to increase skill levels tend to be focussed on the production of animals [91,92] and improving young adult nutrition [94]; thus, their applications may require testing in the horse-owning community. Further research is needed to assess whether basic education can improve the skills of horse owners with regard to improving welfare standards.

#### 4.3.2. Licensing for Regulation

Regulation is discussed in the literature as a possible solution for horse welfare issues but ranks very low in the suggested solutions (Figure 4). This could be due to a lack of consensus amongst stakeholders about what to regulate and how [27] as well as the cost and resources required, such as time and training of regulators, to implement such a programme [40]. However, it should be noted that licensing provides regulatory opportunities. Regulation is the second-highest-ranked licensing success in the literature (Figure 7). A recent study by Liang et al. [95] suggests a need for a security system for animal welfare, using licensing as a regulatory tool. That study was focussed on consumer choice regarding meat consumption; therefore, care should be taken when considering the findings with regard to horse welfare. Also, the concern for security may have been influenced by the COVID-19 pandemic; thus, the experiment would benefit from fresh application to the privately owned horse community. However, increased levels of regulation, potentially through tighter licensing, have also been suggested to safeguard wild animal welfare in the UK [96], suggesting potential for regulation to be promoted across the wider animal welfare sector.

Additionally, increased reporting has been advocated for with regard to horse welfare. A lack of monitoring and reporting is listed as a significant barrier to good horse welfare (Figure 3), and improved reporting is suggested as a solution to horse welfare issues (Figure 4). At the same time, this is identified as a licensing success (Figure 7). A lack of reporting is noted as a problem for determining the welfare status of a population [44], a threat to welfare itself [97] and assessing welfare concerns in sporting scenarios [98]. The database study by Hitchens et al. in 2017 [99] made a compelling case for using an inspection-informed database to monitor animal welfare regionally and internationally. However, this study was reliant on modelling and markers to analyse the data due to the size of the data set, and there was also a risk of inspector bias as they may have been influenced by employer or individual agendas. The paper also notes that some desirable data were lacking in the database, making this method of assessing prevalence currently limited, although promising. The measures of equine welfare in this study do not cover all aspects of welfare; thus, the efficacy of a comprehensive inspection reporting system still needs to be tested. For a reporting system such as this to be implemented, the challenges of cost and centralisation still need to be overcome [97], and then the discussion circles back to how best to assess welfare [44]. However, if it can be implemented correctly, the literature overall appears to support the use of licensing as a reporting tool for the benefit of horse welfare.

A notable challenge when licensing, for any purpose, is ensuring the system works. The most-cited licensing limitation is enforcement (e.g., [27]), followed relatively closely by resources (e.g., [41]) (Figure 8). There are also existing concerns within the literature about the effectiveness of legal systems to enforce current legislation regarding horse welfare [72,100] as well as animal welfare more generally [101]. If resources for animal welfare regulation are already stretched, it could be unreasonable to add to this burden until wider socio-economic issues are resolved. On the other hand, a reduction of animal welfare cases requiring charitable or legal intervention would reduce the cost placed on protecting animal welfare. Thus, it may need to be considered whether short-term investment is necessary for long-term alleviation. There are also concerns that licensing only enforces minimum standards rather than promoting high welfare [102,103], which can, in turn, lead to a lack of trust by the public regarding whether the license is meaningful [100,102]. The considerations for introducing a licensing system in the UK will be explored further below.

#### 4.3.3. Licensing for Education

Being supported by education is listed as one of the top three factors for licensing success (Figure 7). However, there is nothing in the assessed literature to suggest that licensing aids education success. This calls into question the applied usefulness of licensing with regard to horse welfare when used in tandem with education.

Despite this, there are parallels between licensing and education successes and limitations in the assessed literature. National support is highly cited as a required factor for both education, (e.g., [69]) and licensing, (e.g., [104]) to be successful (Figure 5 and Figure 7). Cultural change is also the second-highest cited education success (Figure 5) and ranks in the middle for licensing successes (Figure 7) (e.g., [105]). Additionally, despite some papers advocating that licensing can promote human behaviour change [106], the overall results list a limited influence on human behaviour as a licensing limitation (Figure 8), as with education (Figure 6). It should be noted that the study by Tickle et al. [107], which showed a positive influence on human behaviour from licensing, focussed on a niche group with regard to welfare and licensing, specifically young people involved in hunting in Sweden; thus, the wider literature is more likely to be indicative of the ability of licensing to incite behaviour change as a general principle.

Resource is also a limitation of both education and licensing (Figure 6 and Figure 8). Introducing new licensing requirements dependent on a new education system would be resource heavy and this could prove to be a major barrier to introduction in the UK. By contrast, in France, the licensing and education framework already existed for the competitive riding community [57]; therefore, the new legislation required minimal additional infrastructure or resources. If such a system were to be implemented in the UK, it is possible that a similar gradual approach would be required. Uptake of licensing, or lack thereof, is another highly cited limitation (Figure 8), which again rings true with the previously discussed education limitations. Most studies looking at voluntary licensing schemes are focussed on production animals, (e.g., [93,108]); however, similar problems were found by Collins et al. [97] when exploring horse welfare issues in Ireland. To overcome this, the promotion and introduction of a licensing and/or education system would need to be aligned with the motivations of horse owners in the UK. Finally, it is important to note that a lack of clarity and transparency are both listed as licensing limitations in the literature (Figure 8). A key factor in licensing success would be to make it clear what the license means and what it aims to achieve [109]. In France, the legislation has been stated as combatting horse abuse [16]. However, if individuals who mistreat horses understand it is wrong, as suggested by Collins et al. [100], then, in practice, licensing that targets other welfare issues that do not stem from ignorance or the licensing is being brought in for additional reasons, such as monitoring. Depending on the culture around bureaucracy in each country, public acceptance of a new initiative may be vastly impacted by the openness with which the aims of that initiative are communicated. The considerations for licensing to be accepted in UK society are discussed further later.

### 4.4. Further Considerations for Education

When exploring the possible application of education, thought should be given to how this can be perceived by separate groups within the target population. A key finding from this study was that education needs to be available (Figure 6), which indicates it needs to be delivered in an accessible format and marketed appropriately. Although there have been suggestions to incorporate basic welfare education into the school curriculum, which has been piloted [61,76,80,110], this study is focussed on horse-owner (adult) education; therefore, the applications of this are not discussed in detail.

As horse-owner education is primarily for adults, related education research should be considered. Adults typically take an interest in learning that has immediate relevance to their lives [111]. Knowles’ theory of Adult Learning suggests that learning needs to be collaborative [112], which supports arguments for any form of education for horse owners to have a discursive, communicative element. Additionally, adult learning is self-directed, problem-based and focussed on self-directed learning goals [113]. Adults generally seek education for reasons they perceive as meaningful and will determine the level of responsibility they wish to take [114]. Thus, relating welfare education to horse-owner motivations is a priority, especially to involve those who have the least interest but possibly the greatest need for learning [27].

Gender should not be ignored when considering the findings of this study, as the vast majority of assessed papers involving horse owners have a skew towards women participants. This could be a self-selecting sampling bias or incidental due to recruitment channels. Regardless of the reason, the gender skew in the current research pool has implications for drawing conclusions and next steps. Research has shown that differences still exist between male and female psychology [114]. Notably, Fiedler et al. [115] found that males were less likely to listen to and act on welfare messages. Barriers to education can also be influenced by gender. Women are more concerned about missing work for learning, which may stem from less stable or flexible working arrangements [116]. Women are also more likely to be affected by barriers to education, such as financial constraints and spending time away from family [116], and may be apprehensive of online learning due to ease of use and computer self-efficacy [117]. This suggests that while women may be more open to welfare messages, they feel less able to take on additional learning. Thus, education for horse welfare based on existing research should be designed appropriately; for example, e-learning environments can be more effective for women when they allow for peer-to-peer communication and connectedness [118]. Differences in motivators for education have been also found between genders. For example, females draw motivation from a supportive network of individuals [116]. Additionally, 82.8% of surveyed horse enthusiasts (mostly female) identified other enthusiasts, arguably a close network, as their most important source of information [46]. Although a more recent, smaller survey by Pickering and Hockenhull [74] found that 18.7% of horse owners cited veterinarians as the information source most likely to be consulted compared to 13.1% prioritising other horse owners, the design of that survey asked respondents to make choices in given scenarios and not all participants selected the maximum number of choices available. By contrast, Visser and Wan Wijk-Jansen [46] used open questions about information-seeking behaviour. Thus, the results of the later study do not necessarily indicate general day-to-day information-seeking preferences of horse owners. Therefore, a successful method to motivate female horse owners to learn more about horse welfare could be through popular information sources already feeding into trusted networks. By contrast, if an education initiative is to avoid the risk of excluding men, further research is needed to ensure their views are fully understood regarding barriers to horse welfare.

Horse-owner education should be applied to cater to adult learning styles [113]. This is not necessarily a guaranteed success for welfare education, as some research has found that some pre-existing beliefs may persist if they are rooted in deeply held values [119]; however, the evidence indicates it would increase the likelihood of success. Delivery also needs to be within the national legal framework. In the UK, policy regarding education can vary, sometimes suddenly and with rapid change or reversal, depending on the government in power [120]. Thus, all adult education is voluntary unless required for a particular reason, such as a job role [121,122]. However, a general premise for licensing to mandate education does exist as adults are assessed to be licensed to drive [122]. Therefore, there is scope to introduce a similar system within the UK’s legal framework. However, the present UK policy prioritises learning for young children and upskilling for economic growth [123], meaning that animal welfare education programmes for adults may require significant lobbying to become a high-level concern.

### 4.5. Further Considerations for Licensing

When exploring possibilities for improving monitoring and reporting through licensing, thought should also be given to the national feeling. The UK has been suggested as a low power distance culture, responding positively to public voice opportunities for new projects that directly affect individuals [124]. Other countries with high power distance cultures can be more favourable to new projects decided and implemented without consultation [125]. This suggests that a new licensing system in the UK would be better accepted following public consultation. Interestingly, the survey by Visser and Van Wijk-Jansen [46]—conducted in the Netherlands, another low power distance culture [125]—found that 55.1% of respondents were in favour of governmental regulation for horse welfare. This corresponds with findings by Collins et al. [27] that equine stakeholders believe regulation to be the most effective means for raising welfare standards. However, there remains ambiguity in the existing research about what form this regulation should take.

Furthermore, there could still be resistance to any proposed change regarding licensing due to the culture of bureaucracy in the UK. Whereas countries such as Germany still follow a hierarchical organisation and can be quite accepting of regulation, the UK’s values are based on a more pragmatic approach, where bureaucracy is less present and there is a shift towards more commercial values [126]. While hierarchical bureaucratic systems have their problems, so do their alternatives. Managing through empowerment risks generating outcomes that were meant to be avoided, including de-professionalisation, removal of support networks and the lack of a conflict resolution framework [127]. It could be argued that this has been the approach to horse welfare in the UK. The Animal Welfare Act [128] places responsibility on owners/caregivers to understand and follow the law with regard to the horse’s needs [35], ownership is not governed, regulation of horse registration can be inconsistent [27], and welfare concerns are generally responded to by charitable bodies like the Royal Society for the Prevention of Cruelty to Animals, which often rely on public donations. Perhaps increased bureaucratic intervention is needed to resolve horse welfare concerns in the long term. However, as with education, a regulation system will require resources, and these may not be justified if the system cannot work due to resistance in the public. Bureaucracies need to be impartial as well as efficient for citizen satisfaction [129]; therefore, a new bureaucratic system should be objectively justified to the public to increase the likelihood of effectiveness.

While this study has distinguished between licensing specifically based on educational requirements and licensing for other purposes, these are not mutually exclusive concepts. In fact, Liang et al. [95] suggested providing animal welfare education as well as security systems for animal welfare. This is not the only study to highlight a need for multi-level intervention for animal welfare. May and Previte [66] propose treating animal welfare issues, such as overpopulation, as a network across which social communications (marketing) and education combine to achieve effective regulation. Furtado et al. [77] also propose a holistic approach to achieving human behaviour change, and Douglas et al. [67] argue that the approach to protecting horse welfare needs to be holistic to protect the industry’s social license to operate. Even private horse owners may be affected by a social license to operate, as they are subject to public exposure and could be viewed as representatives of the horse industry as a whole. This approach to welfare marries up with suggestions in the literature that sources of welfare concerns, such as management systems, need to be assessed as a whole, not by parameter [45]. Holistic change also creates the opportunity to apply a collection of strategies with a view to tackling different welfare concerns according to their particular causes. However, one overarching policy and industry change contradicts suggestions to prioritise resources on key welfare issues in the UK [18,42]. The answer to this riddle could be that it is impossible to please all stakeholders, which should not be surprising given the disparities in views explored in Section 4.1. As an alternative, an initial holistic policy change could be implemented, which is then followed by targeted interventions for prioritised welfare concerns to allow for gradual changes to improve horse welfare that may be more practical to implement.

### 4.6. General Discussion

This study has demonstrated the need for sector-wide changes to improve horse welfare. This concurs with a recent report by the RSPCA citing regulation, human behaviour change and increased knowledge as primary focuses for tackling the equine crisis [130]. Overall, the findings of this study indicate that education could increase horse-owner awareness of welfare issues; however, there is a lack of evidence to suggest that licensing directly enhances horse-owner knowledge or ability. It appears that education can improve individual commitment to welfare and acceptance of current ideas while being a popular solution to horse welfare issues. However, to be effective, information sources need to be harmonised, education opportunities need to be accessible to a diverse audience and communication channels optimised for the characteristics of the horse-owning community. However, education also seems to have a limited influence over human behaviour change; thus, it is questionable whether education alone is likely to reduce cases of compromised horse welfare in the UK. Additionally, literature is divided over whether actions to improve horse welfare should prioritise and target the most prominent concerns or apply a holistic blanket approach. Neither method applied in isolation is likely to succeed in resolving all horse welfare concerns; therefore, there is a case for prioritising resources on interventions likely to succeed in combatting the most serious infractions. Licensing can potentially induce cultural change, improve monitoring and aid consumer decision making. However, licensing is most limited by available resources and cost, and there is limited evidence in the literature that licensing can improve welfare standards within a population, although it could aid monitoring and regulation and support or be supported by education. This indicates that licensing horse ownership without education is unlikely to positively impact the welfare state of the UK horse population. However, there is more limited data available for licensing with regard to animal welfare; therefore, more research is needed in this area. Finally, although mandating education and regulating horse ownership has merit in improving the knowledge level within the community, care is needed regarding the bureaucratic culture in the UK compared to other countries. Additionally, education and licensing can both be limited when applied to combat specific welfare concerns, yet it is doubtful that one solution can achieve a blanket improvement in horse welfare in the UK; therefore, other systems of regulation and support may be required.

This study only discussed the most prevalent horse welfare concerns found, as issues with minimal data behind them would be limited in their generalisability to the wider horse population. Additionally, exclusion criteria may have led to some valid yet not-cited papers being overlooked, or health issues mainly studied in commercially owned horses (e.g., racehorses or brood mares) being underrepresented. However, excluded papers that appeared in the database searches were still briefly reviewed, meaning the potential for an important paper to be excluded from this study was minimised. The lack of peer-reviewed publications around horse welfare and licensing, leading to the inclusion of any animal welfare licensing, does highlight the need for this research to be conducted if the use of licensing requirements is to be promoted as part of a solution to horse welfare concerns.

Overall, this study is predominantly theoretical. The next steps for this research would be to investigate the successes and limitations of the newly introduced education and licensing framework in France. In particular, the number of owners previously without a license becoming registered over time, and the number of horse welfare concerns reported in the same time frame compared to before the new legislation. Additionally, data should be collected on horse-owner perception of the new initiative and their knowledge of welfare issues. That data could then be compared with the findings of this study to test the accuracy of these theoretical conclusions and inform future work in this area in relation to UK privately owned horse welfare.

## 5. Conclusions

In conclusion, there are mixed results as to the potential success of using education and licensing to improve privately owned horse welfare in the UK. However, some aspects look promising, such as increased awareness of horse welfare issues with the combination of an education and licensing scheme seeming to have some potential for success. Due to the complex nature of horse welfare issues, educational inventions and licensing systems, careful consideration of format and application needs to be taken before implementation. Further research into these issues would improve chances for success and future improvements in horse welfare.

## Figures and Tables

**Figure 1 animals-15-01037-f001:**
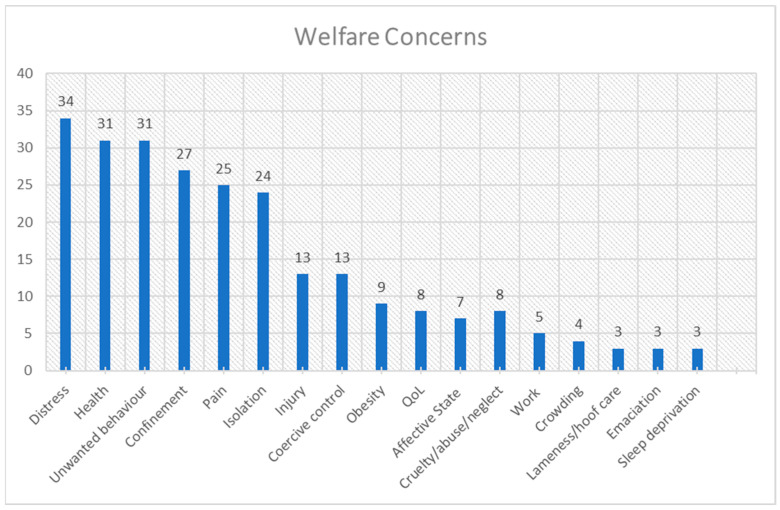
Frequency distribution of welfare concerns (theme 1) from the assessed literature. (QoL = Quality of Life) (Crowding refers to overstocking or too many horses in a given space).

**Figure 2 animals-15-01037-f002:**
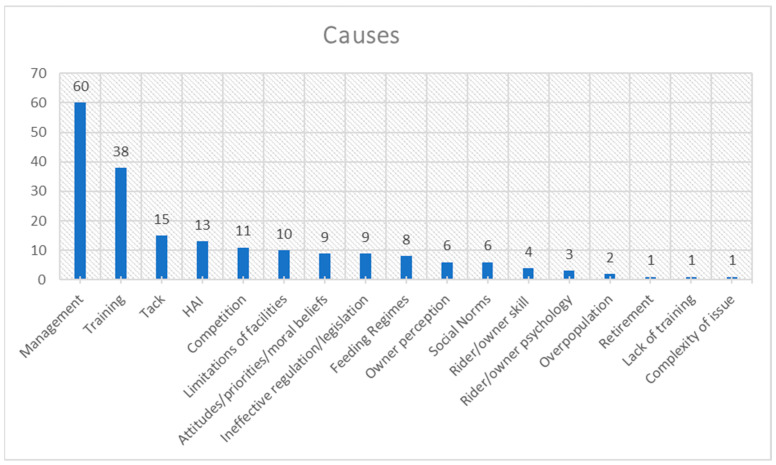
Frequency distribution of causes of poor welfare (theme 2) from the assessed literature. (HAI = Human–Animal Interaction) (Overpopulation refers to too many horses for a given space and can lead to the welfare concern about crowding).

**Figure 3 animals-15-01037-f003:**
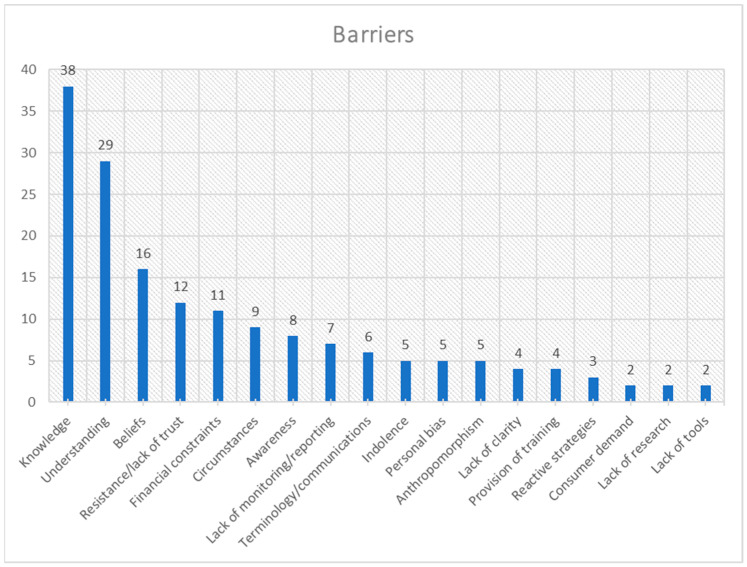
Frequency distribution of barriers to good welfare (theme 3) from the assessed literature. (Indolence refers to a lack of motivation or ‘laziness’ of horse owners to improve equine care knowledge).

**Figure 4 animals-15-01037-f004:**
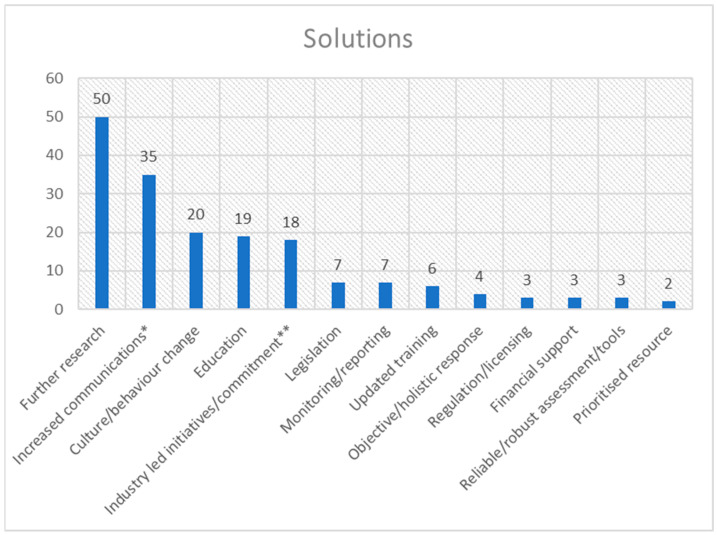
Frequency distribution of solutions to welfare concerns (theme 4) from the assessed literature. [* includes increased awareness. ** includes reviews, example setting, collaboration].

**Figure 5 animals-15-01037-f005:**
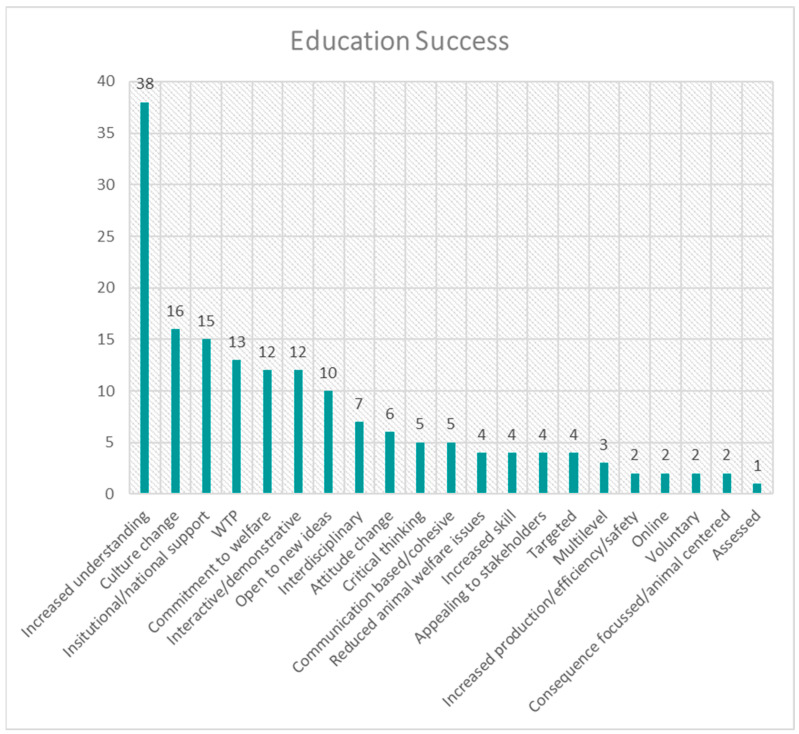
Frequency distribution of successes of education (theme 5) from the assessed literature. (WTP = Willingness/ability To Pay).

**Figure 6 animals-15-01037-f006:**
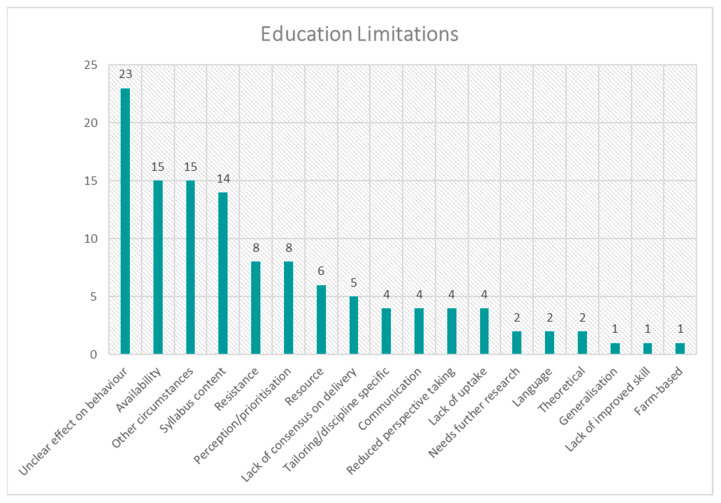
Frequency distribution of limitations of education (theme 6) from the assessed literature.

**Figure 7 animals-15-01037-f007:**
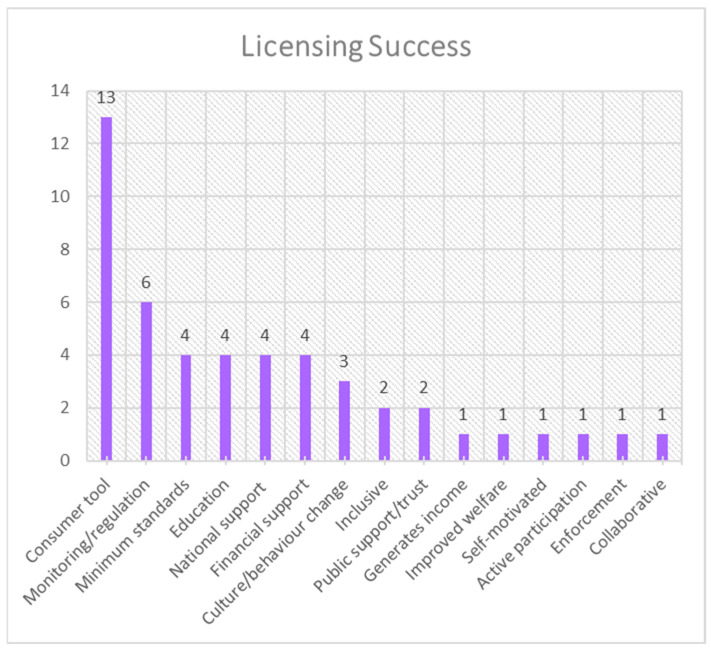
Frequency distribution of successes of licensing (theme 7) from the assessed literature.

**Figure 8 animals-15-01037-f008:**
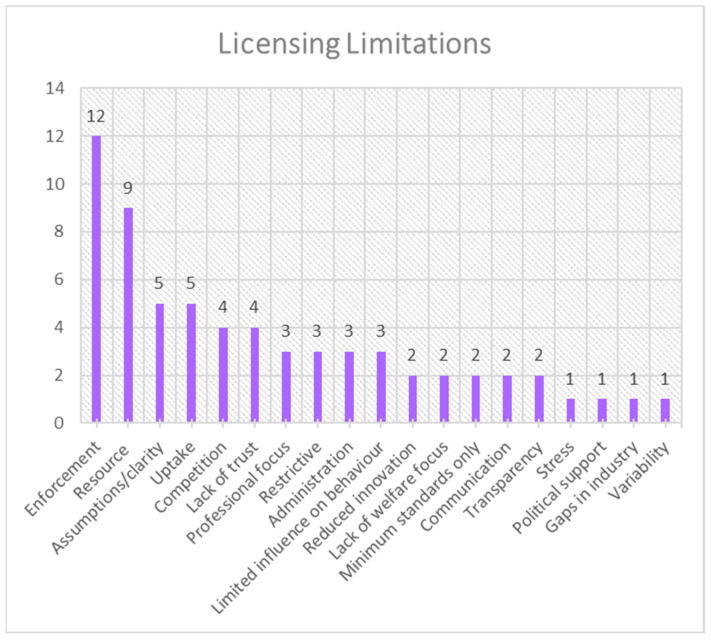
Frequency distribution of limitations of licensing (theme 8) from the assessed literature. (Resource refers to the availability of a program, plan, software, etc., that can be used for licensing.) (Restrictive refers to the limitations or perceptions of limitations the license may place on people’s practices and can lead to resistance to the license.).

**Table 1 animals-15-01037-t001:** Search terms used.

Search Engine	Search 1	Search 2
Web of Science	Horse* OR Equine* (Title) AND welfare OR wellbeing OR wellbeing (title) AND “animal welfare” (Topic) OR Horse* OR Equine* (Abstract) AND welfare OR wellbeing OR wellbeing (Abstract) AND “animal welfare” (Topic)	Education* OR Licensing* (Title) AND “animal welfare” (Topic) OR Education* OR Licensing* (Abstract) AND “animal welfare” (Topic)
Google Scholar	Allintitle: “horse welfare”/“horse wellbeing”/“equine welfare”/“equine wellbeing”	Allintitle: “education animal welfare”/“licensing animal welfare”

**Table 2 animals-15-01037-t002:** Frequency and spread of inputs by theme.

Theme	Input Frequency	Input Variation
1—Welfare Concerns	248	17
2—Causes of Poor Welfare	197	17
3—Barriers to Good Welfare	168	18
4—Solutions	177	13
5—Education Successes	167	21
6—Education Limitations	119	18
7—Licensing Successes	48	15
8—Licensing Limitations	65	19

**Table 3 animals-15-01037-t003:** Most cited inputs per theme (modal theme).

Theme	Highest Input Frequency
1—Welfare Concerns	Distress/stress/fear
2—Causes of Poor Welfare	Management practices
3—Barriers to Good Welfare	Knowledge
4—Solutions to Welfare Issues	Further research
5—Education Successes	Increased knowledge/awareness/ understanding
6—Education Limitations	Unclear effect on behaviour
7—Licensing Successes	Consumer tool
8—Licensing Limitations	Enforcement

## Data Availability

A full list of included papers is available in Appendix B. A list of excluded papers and the full summarized data are available from the author if requested.

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
