# Peer review of "Education and Licensing of Horse Owners: Addressing Poor Horse Welfare in the UK"

_animals, 2025, doi:10.3390/ani15071037_

Round 1
Reviewer 1 Report
Comments and Suggestions for Authors
A really interesting paper, I have added a lot of comments but hopefully none too onerous - I look forward to seeing it published - thanks for writing about such an important topic.
warm regards
T

Author Response
Thank you for your feedback on the manuscript. Please see the attached word document for responses to each comment.

Reviewer 2 Report
Comments and Suggestions for Authors
This review addresses an important and relevant topic, which is challenging in terms of the breadth of factors that can impact on horse welfare. The authors focus on 2 factors, namely education and licensing and provide context for the reasons why these 2 factors are being examined. The main focus is on the UK horse owning population, however reference is made to the developed world on a number of occassions and it is unclear what the relevance is in comparing the frequency of research outside of the UK to that focussed on the UK. I would like the authors to provide more clarity on this.
The review process is robust and the methodology clearly decribed. The results are well presented however comparisons are made to the results of a separate study (table 4) where a different group of stakeholders were asked for their expert opinion. In the current format the relevance of this comparison is not clearly presented and drawing any conclusions around differences and similarities in the perceived welfare issues is questionable when the review focusses on horse owners with specific exclusion of experts and the cited paper focusses specifically on expert opinion. If this is to remain in the paper there needs to be more discussion around the potential reasons for the different perceptions of the main welfare issues affecting horses and how educational material could address this. The discussion does mention briefly the trusted sources of information owners rely on but this could be expanded in relation to the different welfare issues raised by owners and experts and the value experts could bring to raising awareness.
There is a statement in line 318 that is cited but I would question as it states that welfare issues are less likely to be a concern for horses under professional care. In my opinion there would be a similar level of concern but the issues of concern are likely to be different and link closely to social license. Whilst I understand the need to keep the stakeholder group narrow it is important to recognise that welfare issues exist across the sector.
In terms of the defined stakeholder group and the discussion around the value of education it could be helpful to have a bit more detail around the type of private horse owner (on a livery yard vs kept at home) as the issues and strategies to employ for owners on livery yards will be different to those who keep their horses at home
Author Response

(The authors gave the same response as above.)

Reviewer 3 Report
Comments and Suggestions for Authors
Please see attached comments document.

Author Response

(The authors gave the same response as above.)
